# A New Vegetation Index to Detect Periodically Submerged Mangrove Forest Using Single-Tide Sentinel-2 Imagery

**Mingming Jia [1,2]**, **Zongming Wang [1,\*]**, **Chao Wang [2]**, **Dehua Mao [1]** and **Yuanzhi Zhang [3,4]**

1   Key Laboratory of Wetland Ecology and Environment, Northeast Institute of Geography and Agroecology, Chinese Academy of Sciences, No. 4888, Shengbei Street, Changchun 130102, China
2   State Key Laboratory of Information Engineering in Surveying, Mapping and Remote Sensing, Wuhan University, No.129 Luoyu Road, Wuhan 430079, China
3   Center for Housing Innovations, Chinese University of Hong Kong, Shatin, New Territories, Hong Kong
4   Key Lab of Lunar Science and Deep-exploration, National Astronomical Observatories, Chinese Academy of Sciences, Beijing 100101, China
\*   Correspondence: zongmingwang@iga.ac.cn

**Abstract:** Mangrove forests are tropical trees and shrubs that grow in sheltered intertidal zones. Accurate mapping of mangrove forests is a great challenge for remote sensing because mangroves are periodically submerged by tidal floods. Traditionally, multi-tides images were needed to remove the influence of water; however, such images are often unavailable due to rainy climates and uncertain local tidal conditions. Therefore, extracting mangrove forests from a single-tide imagery is of great importance. In this study, reflectance of red-edge bands in Sentinel-2 imagery were utilized to establish a new vegetation index that is sensitive to submerged mangrove forests. Specifically, red and short-wave near infrared bands were used to build a linear baseline; the average reflectance value of four red-edge bands above the baseline is defined as the Mangrove Forest Index (MFI). To evaluate MFI, capabilities of detecting mangrove forests were quantitatively assessed between MFI and four widely used vegetation indices (VIs). Additionally, the practical roles of MFI were validated by applying it to three mangrove forest sites globally. Results showed that: (1) theoretically, Jensen–Shannon divergence demonstrated that a submerged mangrove forest and water pixels have the largest distance in MFI compared to other VIs. In addition, the boxplot showed that all submerged mangrove forests could be separated from the water background in the MFI image. Furthermore, in the MFI image, to separate mangrove forests and water, the threshold is a constant that is equal to zero. (2) Practically, after applying the MFI to three global sites, 99–102% of submerged mangrove forests were successfully extracted by MFI. Although there are still some uncertainties and limitations, the MFI offers great benefits in accurately mapping mangrove forests as well as other coastal and aquatic vegetation worldwide.

**Keywords:** Sentinel-2 MultiSpectral Instrument (MSI); red-edge band; aquatic vegetation; tidal condition; vegetation index; coastal vegetation

## 1. Introduction

Mangrove forest are highly productive ecosystems with significant ecological and socio- economic importance in the world [1,2]. However, over the past century, these forests have declined at an alarming rate that is more rapid than that of inland tropical forests [3]. Therefore, there is an emerging demand for conservation and restoration efforts in mangrove forests. Obtaining accurate information

regarding the current and past acreage and condition of mangrove forests is essential for efficient management of these ecosystems and for policy- and decision-making processes [4,5].

Located in intertidal zones, mangrove forests are often inaccessible for traditional field surveys. For decades, remote sensing has been widely used to monitor the distribution of mangrove forests, yet accurate and timely interpretation of the relatively small patches has been rare, due to the lack of full consideration of tidal conditions [6–9]. Mangrove forests located near the shoreline are periodically submerged by tides, especially in regions with high tide fluctuations and lower mangrove shrubs [9,10]. Ideally, it is better to use images acquired during low tides; however, such data are difficult to obtain, due to uncertainties of local instantaneous tidal conditions during predetermined times that satellites pass over [11,12]. For a long time, numerous studies have pointed out that tides may seriously influence remote sensing results of mangrove forests, yet, solutions were not reported until the past two years [10,13,14]. However, all these studies used multi-tides (multi-date) images; therefore, we have one concern: if multi-tides images are not available due to rainy climates and uncertain local tide conditions, how could we accurately map mangrove forests by a single-date image?

Over the last two decades, remote sensing of submerged and emerged aquatic vegetation has been widely studied [15,16]. Hyperspectral image with numerous narrow and contiguous bands is reliable for studying aquatic vegetation and is able to detect the biophysical properties of vegetation efficiently [16–19]. However, there is no freely available satellite hyperspectral data in recent years, and airborne applications are exorbitantly expensive and only cover a very small spatial extent. Landsat images with moderate spatial resolution of 30–60 m have been widely used for mapping aquatic vegetation [20–24]. Yet, Landsat only has one band in the spectral region of near infrared (760–900 nm), which may become less sensitive as water depth increases [25]. The MODIS (Moderate Resolution Imaging Spectroradiometer) and AVHRR (Advanced Very High Resolution Radiometer) are publicly available with high spectral resolution but coarse spatial resolution (250–1100 m, respectively), making them unsuitable for mangrove detection [9]. In contrast, the Sentinel-2 MultiSpectral Instrument (MSI) sensor has a 10–20 m spatial resolution and five bands in near infrared region, which provides opportunities to conduct quick, robust, and efficient monitoring of submerged mangrove forests.

For years, numerous methods were utilized to map mangrove forests as well as other aquatic vegetation from remote sensing imagery, ranging from pixel to object-oriented approach, and manual to unsupervised methods [9,14,26–28]. Recently, machine-learning algorithms such as random forest, neural network, and support vector machine provide promising accuracy in mangrove forests extraction [10,13,29]. As it is hard to locate representative training samples due to uncertain tidal conditions, it is relatively hard to apply these methods to extract submerged mangrove forests from a single-date image. Vegetation indices (VIs), which are mathematically determined based on the spectral characteristics of vegetation, have been proven efficient in monitoring vegetation from space [30]. The Normalized Difference Vegetation Index (NDVI) is the most commonly used index in global vegetation studies (e.g., [30,31]). The Land Surface Water Index (LSWI) and the Modified Normalized Difference Water Index (MNDWI) were proposed and widely used for mapping surface water [32–34]. Given that these indices are established based on differences between two bands, they are insensitive to small variations of the reflectance of submerged mangrove forests and water background [14]. Furthermore, it is hard to decide thresholds that distinguish submerged mangroves and water. With more bands, several VIs were built based on a baseline theory, such as Maximum Chlorophyll Index (MCI; [35]), the Floating Algae Index (FAI; [36]), and the Floating Vegetation Index (FVI; [17]). However, these indices were defined to extract floating vegetation (above water surface) from water, not submerged vegetation. Meanwhile, bands used to build MCI and FVI did not exist in Sentinel MSI image.

Thus, the objective of this study is to develop a new vegetation index, called the Mangrove Forest Index (MFI), which is capable to map the distribution of mangroves based on a single date MSI image. Then, we will compare MFI with other widely used VIs to validate MFI's capabilities in detecting submerged mangrove forests from water background. Additionally, MFI will be applied to three sites

of typical mangrove forests worldwide; the practical roles of mapping mangrove forests during local high-tide conditions will also be discussed.

## 2. Materials and Methods

### 2.1. Sentinel-2 Imagery

Sentinel-2, a European Space Agency (ESA) land-monitoring mission, has two matching satellites that provide high-resolution optical imagery. Sentinel-2A and Sentinel-2B carry the MultiSpectral Instrument (MSI) and were successfully launched in June 2015 and March 2017 respectively and provide important means to augment earth observation capabilities [37]. These satellites revisit the same location every 2 to 5 days. The MSI sensor offers 13 spectral bands, with four bands at 10 m, six bands at 20 m, and three bands at a 60 m spatial resolution (Table 1) and offers a wide range of earth observation applications [38].

In this study, Sentinel-2 MSI images were downloaded from European Space Agency Sentinels Scientific Data Hub; the images were preprocessed with geometric and radiometric corrections at sub-pixel accuracy. Then, atmospheric correction (converting top-of-atmosphere reflectance into top-of-canopy reflectance) was performed by the tool of SEN2COR (version 2.05.05), which was available in the Sentinel Application Platform (SNAP) toolbox [39,40]. In order to standardize different spatial resolutions of bands in MSI images, we excluded bands with a spatial resolution of 60 m (Band 1, Band 9, and Band 10). After atmospheric correction, all remaining bands were resampled to a pixel size of 20 m × 20 m.

**Table 1.** General characteristics of the Sentinal-2 MultiSpectral Instrument (MSI) sensor.

| MSI Band | Band Name | Wavelength (Central, nm) | Spectral Width (nm) | Spatial Resolution (m) |
|---|---|---|---|---|
| B1 | Aerosols | 443 | 20 | 60 |
| B2 | Blue | 490 | 65 | 10 |
| B3 | Green | 560 | 35 | 10 |
| B4 | Red | 665 | 30 | 10 |
| B5 | Vegetation red-edge | 705 | 15 | 20 |
| B6 | Vegetation red-edge | 740 | 15 | 20 |
| B7 | Vegetation red-edge | 783 | 20 | 20 |
| B8 | Near infrared | 842 | 115 | 10 |
| B8A | Vegetation red-edge | 865 | 20 | 20 |
| B9 | Water-vapor | 945 | 20 | 60 |
| B10 | Cirrus | 1380 | 30 | 60 |
| B11 | Shortwave-infrared reflectance (SWIR)1 | 1610 | 90 | 20 |
| B12 | SWIR2 | 2190 | 180 | 20 |

### 2.2. Study Area

The study area, Zhenzhu Harbor (21°28′–21°42′N and 108°00′–108°20′N), is located in Guangxi Province, China, in the southwest portion of mainland China and the north region of Tonkin Gulf (Figure 1). In Zhenzhu Harbor, mangrove forests are mainly composed of four communities: *Comm. A. marina, Comm. A. corniculatum, Comm. A. marina–A. corniculatum,* and *Comm. K. candel–A. corniculatum.* The tides in the coast of Zhenzhu Harbor are diurnal, with an average tidal range of 2.24 m, and mangrove forests here are primarily younger shrubs with an average height of less than 3 m [7]. Therefore, the Zhenzhu Harbor is a typical area for the study of submerged mangrove forests, due to a large area of pioneer mangrove trees and shrubs that would be entirely submerged during high tides (Figure 1A,B). Information of MSI images we selected are shown in Table 2.

Additionally, a field survey of Zhenzhu Harbor was conducted during April 2017, in which 408 ground truth samples were collected including samples of mangrove forest, open water, and other land cover.

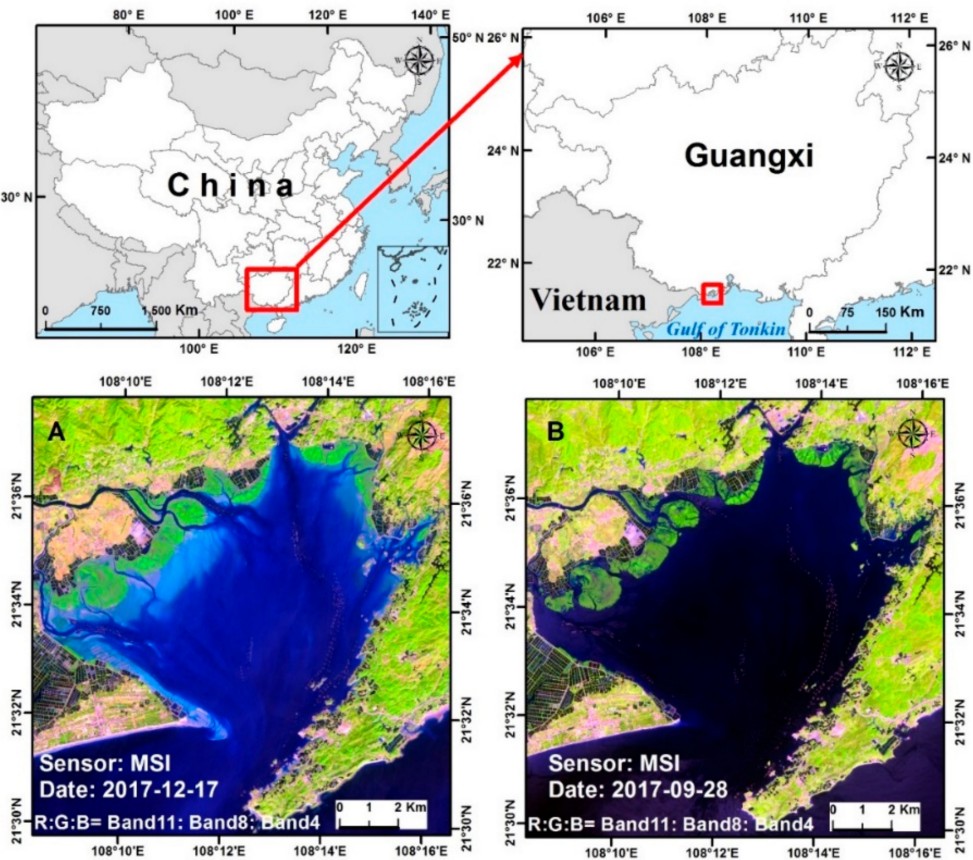

**Figure 1.** Snapshots of low- and high-tide Sentinel MSI images of study area ((**A**) during local low tide all mangrove forests were emerged; (**B**) during local high tide some of the mangrove forests were submerged).

**Table 2.** Description of satellite data, including the path, row, date, time of acquisition, and tide level of the nearest tide station (Fangcheng Harbor Station, 108°14'E, 21°28'N).

| Sensor | Path | Row | Date | Time (hh:mm) | Tide Height (m) |
|--------|------|-----|------|--------------|-----------------|
| MSI | 205 | 118 | 2017-12-17 | 11:23 | −0.9 |
| MSI | 205 | 118 | 2017-09-28 | 16:37 | 1.8 |

### 2.3. Build a Reference Map

Ground surveys were conducted along the coasts of Zhenzhu Harbor in November 2017. The location of each sampling point was measured using a global positioning system (GPS), with an error less than 1 m. The observations collected in the surveys contained 85 mangrove points and 81 water points. A vector file of ground survey points with the attributes of location (longitude and latitude), land cover types, and photos was created with ArcGIS.

The spectral curves of water, submerged mangrove forests, and emerged mangrove forests were extracted from images. The workflow of discriminating these classes is shown in Figure 2. A reference map was built based on object-oriented methods and visual interpretation.

The description of the object-oriented method can be found in Harayama and Jaquet [41]. The eCognition Developer 9.0, an image analysis program, was used to conduct object-oriented classification. Visual interpretation was performed to classify objects as either mangrove forests or water. To facilitate

visual interpretation, a false color composite of MSI Bands 11 (centered 1610 nm), Band 8 (centered 842 nm), and Band 4 (centered 665 nm) was generated. This band combination that was deemed the best for detecting mangroves which appears dark green color [42]. Furthermore, in order to conduct the adjustment in a robust manner, visual interpretation was performed by an experienced remote sensing expert who was familiar with this area. First, we identified mangrove forest and surrounding water from the low-tide MSI image. Subsequently, a confusion matrix was generated using the independent ground-truth samples described in Section 2.2. With this matrix, we achieved an overall classification accuracy of 97% with a Kappa coefficient of 0.92, which indicated excellent agreements between our mapping results and ground-truth data. Therefore, mangrove forests identified with this method were assumed to cover entire area of local mangrove forests. Second, using the same techniques as above, mangrove forests were identified from the high-tide MSI image. Finally, the extent of the submerged mangrove was determined by subtracting the high-tide from the low-tide mangrove forest map. Figure 3 shows the distributions of emerged mangrove forests and submerged mangrove forests in the high-tidal MSI image.

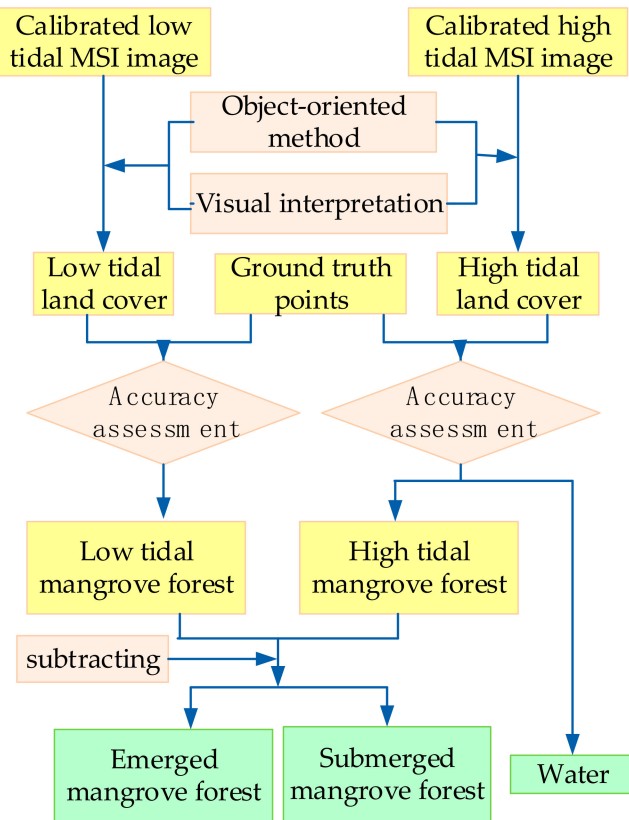

**Figure 2.** Work flow for identifying submerge mangrove forests.

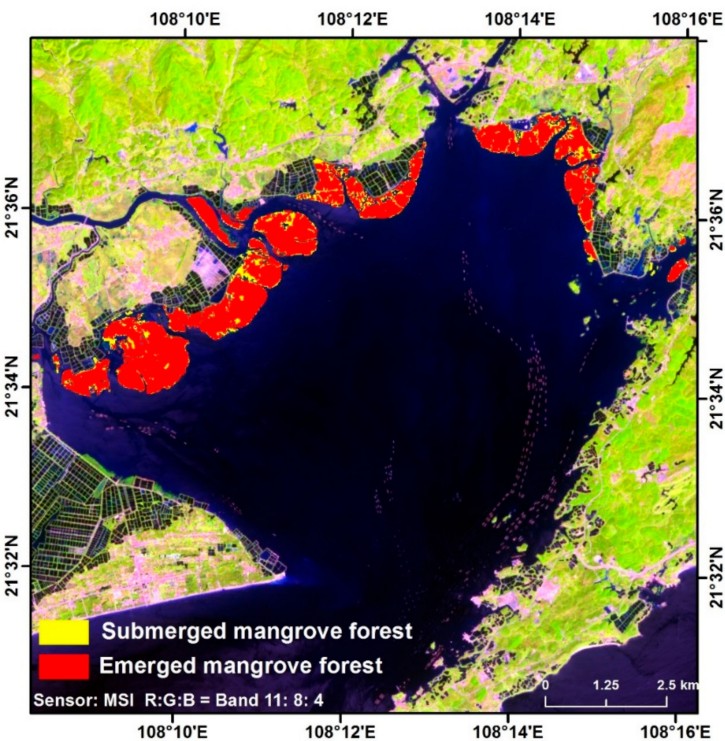

**Figure 3.** Distribution of emerged and submerged mangrove forests in high-tide MSI image.

*2.4. Theories*

Figure 4 shows the field measurements of spectral curves of the water, emerged vegetation, and submerged vegetation, generated by Chen et al. (2018; Figure 4A) and Visser et al. (2015; Figure 4B) [15,25]. As normal green plants, vegetation above the water surface showed high reflectance in the spectral region of 770–890 nm. Waterbody is characterized by low reflectance in near infrared at 700 nm while emerged mangrove has a relatively high reflectance, which make them separable from each other. However, when mangroves are submerged under water, the reflectance is largely reduced, therefore, it is difficult to distinguish submerged vegetation from waterbody [15,25,43]. As measured, when submerged vegetation are 43–51 cm below clear water, the NDVI value was close to zero, which means no differences were observed between the red band and NIR band [44].

However, by careful inspection of the spectral curves shown in Figure 4A,B, two reflectance peaks ranging from approximately 690–740 nm and 810–830 nm were found, even for the curves of vegetation located 40 cm below the water surface. These peaks result from the competing effects of the chlorophyll reflectance plateau and the absorption effects of water located within submerged vegetation and the surrounding water background. However, traditional multispectral satellite sensors could not capture these reflectance peaks. Fortunately, the MSI sensor has five channels that cover these regions. Figure 5 shows the typical spectral curves of water (WB), emerged mangrove forest (EMF), and submerged mangrove forest (SMF) that are observed on the MSI image. As shown in Figure 5, the emerged and shallow submerged mangrove forest pixels demonstrated a strong reflection in the region of 660–900 nm, the absorption valleys appeared in bands 4 (centered wavelength 665 nm) and 12 (centered 2160 nm). For the submerged mangrove forest (b) curves, a small reflectance peak appeared in the 660–900 nm region; the absorption valley also appeared in bands 4 (centered 665 nm) and 12 (centered 2160 nm). The reflectance of the water pixels shows a continuous decreasing tendency beginning with band 4 (central wavelength 665 nm). Therefore, comparing submerged curves to water curves, the higher reflectance in bands 5 (centered 705 nm), 6 (centered 740 nm), 7 (centered 783 nm), 8 (centered 842 nm), and 8A (centered 865 nm) could be used to distinguish submerged mangrove forests from the water background.

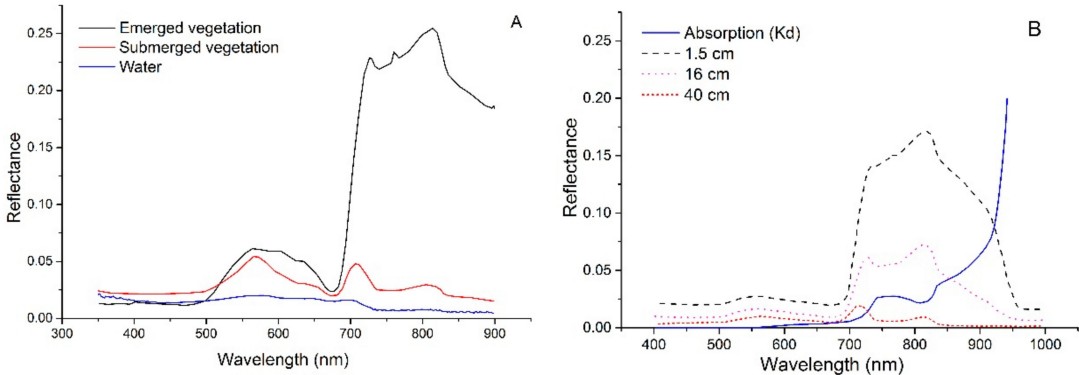

**Figure 4.** The spectral curves of water, emerged, and submerged vegetation, as well as the absorption coefficients of water (cm$^{-1}$). ((**A**) field-measurement [15]; (**B**) field-measured of submerged vegetation's reflectance at 1.5, 16, and 40 cm below water surface [25]).

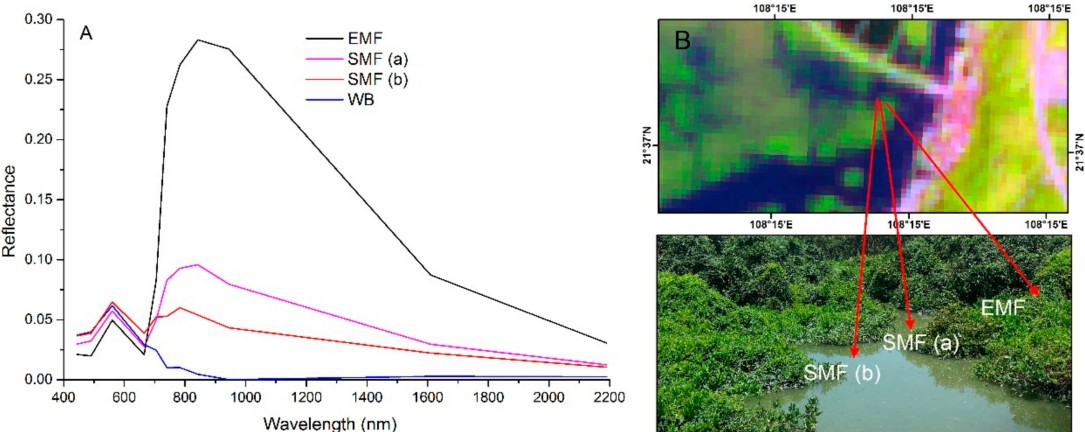

**Figure 5.** (**A**) Typical spectral curves of emerged (EMF), submerged mangrove forests (SMF) and water (WB) in Sentinel-2A MSI image. (**B**) EMF and SMF forests in Sentinel MSI image and field photo. (**a**) Represents shallow submerged mangrove forests (0–30 cm), (**b**) Represents deep submerged mangrove forests (30–60 cm).

### 2.5. Existing Vegetation Indices

Previously, NDVI (Equation 1), LSWI (Equation 2), and MNDWI (Equation 3) and FAI (Equation 4) were used in detecting vegetation from water bodies [33,34,45].

$$\text{NDVI} = \frac{\rho_{\text{NIR}} - \rho_{\text{Red}}}{\rho_{\text{NIR}} + \rho_{\text{Red}}} \tag{1}$$

$$\text{LSWI} = \frac{\rho_{\text{NIR}} - \rho_{\text{SWIR}}}{\rho_{\text{NIR}} + \rho_{\text{SWIR}}} \tag{2}$$

$$\text{MNDWI} = \frac{\rho_{\text{Green}} - \rho_{\text{SWIR}}}{\rho_{\text{Green}} + \rho_{\text{SWIR}}} \tag{3}$$

$$\text{FAI} = \{\rho_{860} - [\rho_{1240} + (\rho_{660-1240}) \times (1240 - 860)/(1240 - 660)]\} \tag{4}$$

where $\rho_{\text{Green}}$, $\rho_{\text{Red}}$, $\rho_{\text{NIR}}$, and $\rho_{\text{SWIR}}$ are the reflectance of the green, red, NIR, and SWIR, respectively. However, these indices are not suitable for discerning submerged vegetation from water bodies, because there are no obvious reflectance differences in bands green, red, and SWIR between submerged vegetation and water bodies (Figures 4 and 5).

### 2.6. Formulation of MFI

For this study, according to the analysis in Section 2.4, the absorption valleys in band 4 (centered 665 nm) and band 12 (centered 2190 nm) could be used to form a baseline (Figure 6). In order to enhance the stability of differences between submerged mangrove forests and the water background, the average value of reflectance of band 5 (centered 705 nm), band 6 (centered 740 nm), band 7 (centered 783 nm), and band 8A (centered 865 nm) above the baseline, is defined as MFI. Band 8 was excluded because Bands 7 (centered 783 nm) and 8A (centered 865 nm) covered most of its spectra, and its spectral range overlaps with the water absorption region. The mathematical formulation is

$$\text{MFI} = \left[ (\rho_{\lambda 1} - \rho_{B\lambda 1)+} (\rho_{\lambda 2} - \rho_{B\lambda 2)+} (\rho_{\lambda 3} - \rho_{B\lambda 3)+} (\rho_{\lambda 4} - \rho_{B\lambda 4}) \right]/4 \tag{5}$$

$$\rho_{B\lambda i} = \rho_{2190} + (\rho_{665} - \rho_{2190}) \times (2190 - \lambda i)/(2190 - 665) \tag{6}$$

where the $\rho_{\lambda}$ is the reflectance of the band center of $\lambda$, and $i$ ranged from 1 to 4; $\lambda 1$, $\lambda 2$, $\lambda 3$, $\lambda 4$ represent the center wavelengths at 705, 740, 783 and 865 nm, respectively. $\rho_{B\lambda i}$ is the baseline reflectance in $\lambda i$. $\rho_{665}$ and $\rho_{2190}$ are the reflectance of band 4 (centered at 665 nm) and 12 (centered at 2190 nm), respectively. Pixels with an MFI value above 0 are recognized as mangrove forests.

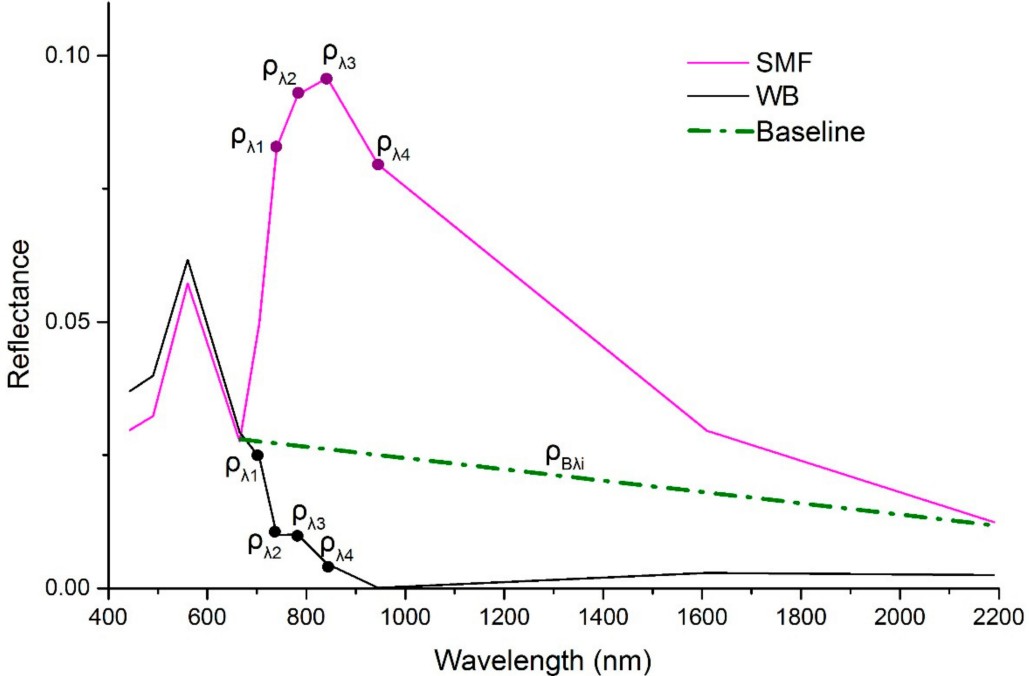

**Figure 6.** Baseline theory of establishing Mangrove Forest Index (MFI), including reflectance of submerged mangrove forest and water.

### 2.7. Quantitative Comparison between MFI and Other VIs

In this study, Jensen–Shannon divergence (JSD)—a measure of distance between a finite number of distributions—was adopted to compare sensitivities of MFI and other VIs. The JSD ($D$) quantifies the difference between two or more probability distributions. In this study, it was used to compare the differences between submerged mangrove forests and water pixels in different VI images. The $D$ value, which was calculated in MATLAB, is defined as follows: let $p^{(1)} \equiv (p_1^{(1)}, p_2^{(1)}, \ldots, p_k^{(1)})$ and $p^{(2)} \equiv \left( p_1^{(2)}, p_2^{(2)}, \ldots, p_k^{(2)} \right)$ denote two probability distributions satisfying the usual constraints $\sum_{i=1}^{k} p_i^{(j)} = 1$ and $0 \leq p_i^{(j)} \leq 1$ for all $i = 1, 2, \ldots, k$ and $j = 1, 2$; and let $\pi^{(1)}$ and $\pi^{(2)}$ denote the weights of the distributions $p^{(1)}$ and $p^{(2)}$, satisfying the constraints $\pi^{(1)} + \pi^{(2)} = 1$ and $0 \leq \pi^{(j)} \leq 1$. Then the

Jensen–Shannon divergence $D$ between the probability distributions $p^{(1)}$ and $p^{(2)}$ with weights $\pi^{(1)}$ and $\pi^{(2)}$ is defined by [46]:

$$D\left[p^{(1)},\, p^{(2)}\right] \equiv H\left[\pi^{(1)}p^{(1)} + \pi^{(2)}p^{(2)}\right] - \left(\pi^{(1)}H\left[p^{(1)}\right] + \pi^{(2)}H\left[p^{(2)}\right]\right) \tag{7}$$

where

$$H[p] = -\sum_{i=1}^{k} p_i \log_2 p_i \tag{8}$$

denotes the Shannon entropy of the probability distribution $p \equiv (p_1, p_2, \ldots p_k)$. $D$ ranging from 0–1, 0 means no difference between distributions, 1 means the distributions are completely different.

## 3. Results

### 3.1. Quantitative Comparison of MFI, FAI, NDVI, LSWI, and MNDWI

To compare the ability to distinguish submerged mangrove forests from the water background, the VIs values of all submerged mangrove forests (in total 12,001 pixels extracted in Section 2.3), and 22,668 pixels of water in the high-tidal MSI image (acquired on 28 September 2017) were calculated. Additionally, to make the different VIs comparable, the MFI and FAI were calculated to 10 times of their original values. The boxplots of submerged mangrove forests and water pixels are shown in Figure 7.

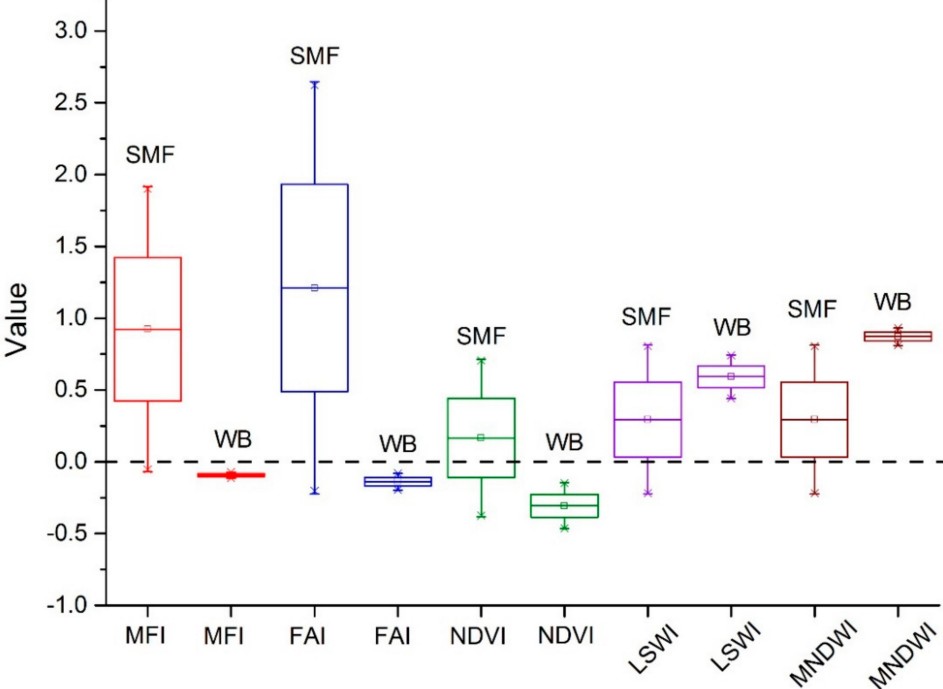

**Figure 7.** Boxplot of different index values over submerged mangrove forest pixels and water pixels (MFI and Floating Algae Index (FAI) are 10 times their original value. SMF means submerged mangrove forest, WB means Water Body. The horizontal axis represents different indices).

As shown in Figure 7, all submerged mangrove forest pixels have higher MFI values than those of the water background, the minimum value of submerged mangrove forests is equal to the maximal value of water. Most of the water pixels are confused with submerged mangrove forests in FAI, NDVI, and NDWI image. According to our calculations, the $D$ values of MFI, FAI, NDVI, LSWI, and MNDWI are 0.209, 0.077, 0.012, 0.003, and 0.121, respectively, which means pixels of submerged mangrove forests and water are better separated in an MFI image than other VIs.

## 3.2. Evaluation of MFI at Different Mangrove Forests around the World

Globally, three selected mangrove forest sites were chosen to demonstrate the practical utility of our newly formed index (MFI) in distinguishing mangrove forests from water background. They are (a) Zhenzhu Harbor, Guangxi, China, (b) Dalhousie Island, Sundarbans, India, (c) Baia do Arraial, Amazon Coast, Brazil. Locations are shown in Figure 8.

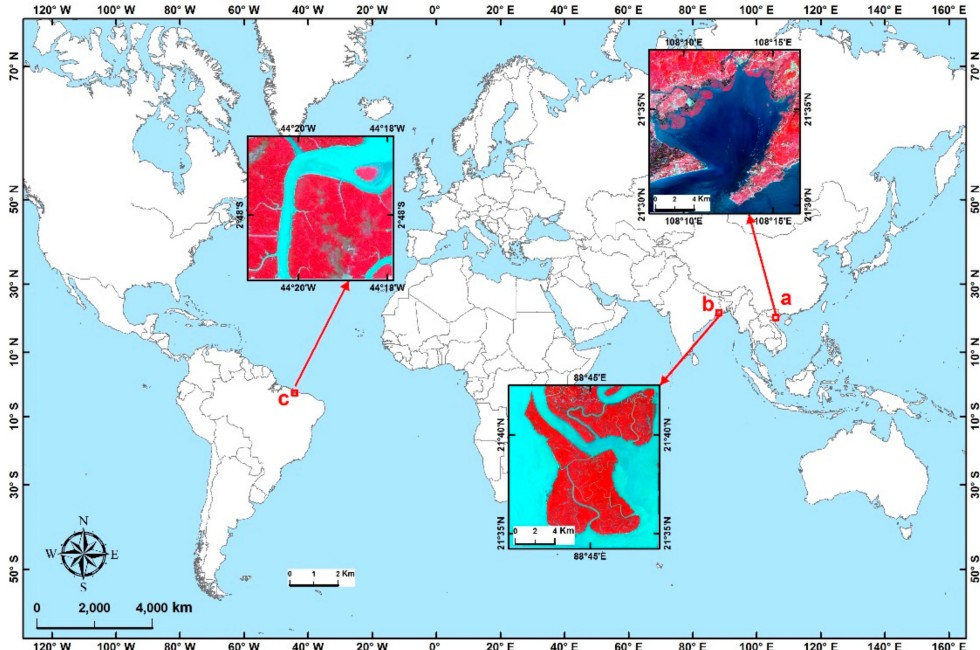

**Figure 8.** Global study sites of mangrove forests. (Displayed imagery: R:G:B = Sentinel MSI Band 8A: 4:3. (**a**) Zhenzhu Harbor, Guangxi, China; (**b**) Dalhousie Island, Sundarbans, India; (**c**) Baia do Arraial, Amazon Coast, Brazil).

Classification accuracy assessment is essential for validating the performance of the MFI index. In this study, a table containing the overall accuracy, user's accuracy, producer's accuracy, and the Kappa coefficient of each site were presented in Table 3. In Zhenzhu Harbor, the validation samples were collected from field survey. In Dalhousie Island and Baja do Arraial, validation samples were randomly selected from Google Earth high-resolution images.

**Table 3.** Confusion matrix for worldwide study sites of mangrove forests, including overall accuracy, producer's accuracy, user's accuracy, and Kappa coefficient.

| Zhenzhu Harbor Land Cover | Classification Results | | |
|---|---|---|---|
| | **Mangrove** | **Water** | **Producer's Accuracy** |
| Mangrove | 82 | 3 | 96.4% |
| Water | 2 | 79 | 97.5% |
| User's accuracy | 97.6% | 96.3% | – |
| Overall accuracy | 97.0% | Kappa coefficient | 0.94 |
| **Dalhousie Island** | Mangrove | Water | Producer's accuracy |
| Mangrove | 52 | 1 | 98.1% |
| Water | 2 | 39 | 95.1% |
| User's accuracy | 96.2% | 97.5% | – |
| Overall accuracy | 96.8% | Kappa coefficient | 0.93 |
| **Baja do Arraial** | Mangrove | Water | Producer's accuracy |
| Mangrove | 30 | 3 | 90.9% |
| Water | 3 | 36 | 92.3% |
| User's accuracy | 90.9% | 92.3% | – |
| Overall accuracy | 91.7% | Kappa coefficient | 0.83 |

### 3.2.1. Zhenzhu Harbor, Guangxi, China

MFI was applied to the high-tidal Sentinel MSI image (acquired 2017-09-28). Figure 9 shows the MFI image (Figure 9A), mangrove forest distribution classified from MFI image (Figure 9B), and reference map (Figure 9C) derived from low-tide Sentinel MSI images (described in Section 2.3). According to the results shown in the reference map (Figure 9C), the total area of mangrove forests was 856.30 ha, with 107.05 ha of submerged and 749.25 ha of emerged mangrove forests. In Figure 9B, the total area of mangrove forest we classified from the MFI image was 849.4 ha, which means 99% of the mangrove forest pixels were successfully extracted from water background by the MFI. According to Table 3, the overall accuracy of this mangrove map is 97% with a Kappa coefficient of 0.94. In Zhenzhu Harbor, the MFI value of emerged mangrove forests, submerged mangrove forests, and water pixels range from 0.19 to 0.30, −0.01 to 0.18, and −0.2 to −0.01.

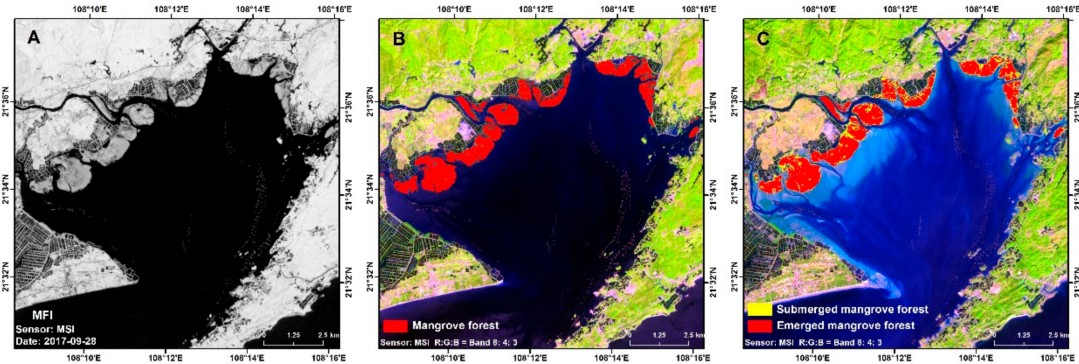

**Figure 9.** Apply the MFI to extract mangrove forests in Zhenzhu Harbor, Guangxi, China. (**A**) MFI image, (**B**) mangrove forests extracted from MFI image, and (**C**) reference map.

### 3.2.2. Dalhousie Island, Sundarbans, India

Sundarbans has the biggest patch of mangrove forest flourishing on the world's largest delta (Ganga–Bramhaputra–Meghna Delta; [47]). The tidal amplitude within the estuary ranges from 3.5 to 4 m, with seasonal variation between 1 and 6 m; mangrove forests are periodically submerged during high tide [48,49]. In this study, Dalhousie Island (located in the southern part of Sundarbans, India) was chosen as a typical area to validate the performance of the MFI. Figure 10 shows a local high-tidal MSI image (Figure 10A, captured on 2016-10-17), the MFI-derived image (Figure 10B), and

ground-truth images obtained by Google Earth snapshot (Figure 10a–c). As shown in Figure 10A, in mangrove swamps, a number of patches seem similar to seawater. In Figure 10a–c, although these patches show white tones, they are lower and sparser mangrove forests that are intermittently flooded by tides. Fortunately, these patches have positive values in the MFI image (Figure 10B). According to Mondal and Saha (2018), Dalhousie Island had 5950 ha of mangrove forests on 2015-08-03 [50]. In the MFI image, the extent of mangrove extracted by the MFI is 6105 ha (pixels with positive MFI values), accounting for 102% of Mondal and Saha's result. Although the MSI image was captured during high tide, almost all the local mangrove forests were detected by the MFI. According to Table 3, the overall accuracy of this mangrove map is 96.8%, with a Kappa coefficient of 0.93. In Dalhousie Island, the MFI value of emerged mangrove forests, submerged mangrove forests, and water pixels range from 0.11 to 0.25, 0 to 0.10, and −0.03 to 0.

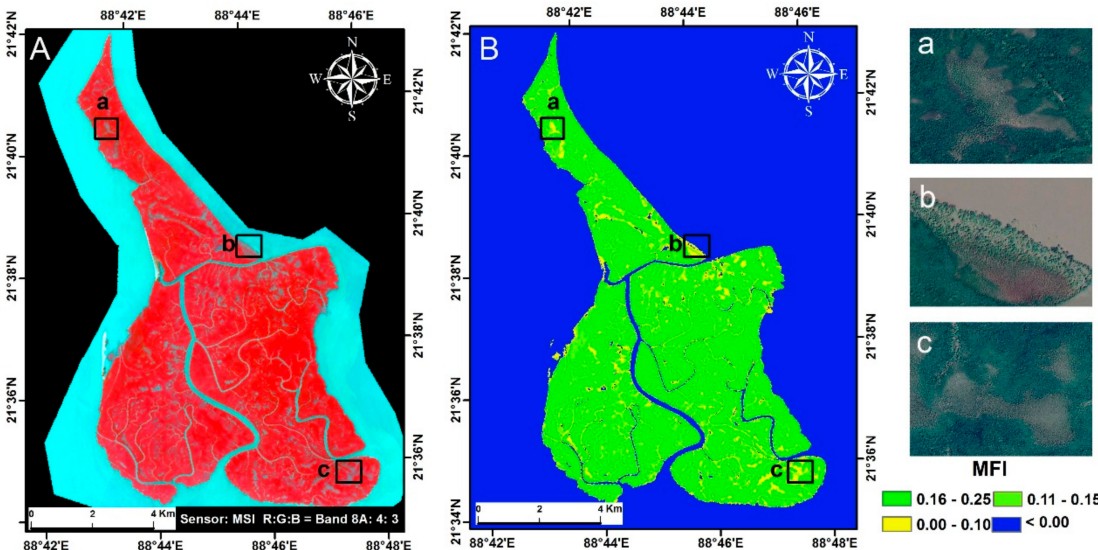

**Figure 10.** Apply the MFI to extract mangrove forests in Dalhousie Island, Sundarbans, India. (**A**) Sentinel MSI image (Band combination: R:G:B = 8A: 4: 3); (**B**) Sentinel MSI-based MFI image; (**a–c**): Google Earth snapshot.

### 3.2.3. Baia do Arraial, Amazon Coast, Brazil

Baia do Arraial is located along the south coasts of São Luís city, Brazil (Figure 11). The coastal zone of São Luís is dominated by a semidiurnal tide; the high energy causes a maximum tidal height of 8 m during the equinoctial spring tide. Therefore, numerous mangrove trees and shrubs would be submerged during high tides. As shown in Figure 11A, on 2018-06-14, patches in the northwest and the middle of mangrove swamps were submerged; fortunately, these mangrove forests showed positive values in Figure 11B, and the snapshot of Google Earth images confirmed that these places were occupied by low mangrove forests. Therefore, we concluded that submerged mangrove forests in Baia do Arraial could be detected by the MFI. According to Table 3, the overall accuracy of this mangrove map is 91.7%, with a Kappa coefficient of 0.83. In Baia do Arraial, the MFI values of emerged mangrove forests, submerged mangrove forests, and water pixels range from 0.09 to 0.25, 0 to 0.09, and −0.06 to 0.

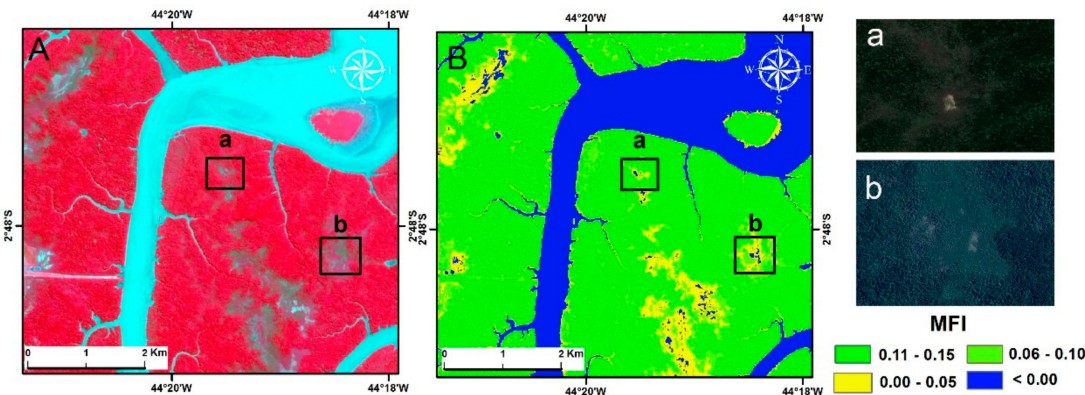

**Figure 11.** Apply the MFI to extract mangrove forests in Baia do Arraial, Amazon Coast, Brazil. (**A**) Sentinel MSI image (band combination: R:G:B = 8A: 4: 3), (**B**) Sentinel MSI based MFI image, (**a**,**b**): Google Earth snapshot.

## 4. Discussion

### 4.1. Advantages and Potential Applications of MFI

Located along intertidal zones, mangrove forests are always relatively small patches; therefore, misclassification of a small area would greatly affect mapping results. The lack of full consideration of tidal conditions would cause misclassification between mangrove forests and water background. To accurately map and manage mangrove forests, in this study, we attempt to extract all mangrove forests during local high tides. Undeniably, using VIs to extract mangrove forests is not new. All the commonly used indices are applicable to detecting emerged mangrove forests. However, during high tides, according to our statistics in Figure 7, in LSWI, MNDWI, NDVI, and FAI images 27%, 8%, 19%, and 5% of submerged pixels were mixed with water background. Furthermore, based on the result of Jensen–Shannon divergence, MFI greatly increased the distance of submerged mangrove forests and water. However, in the MFI image, nearly all the submerged pixels were completely separated from the water background. Furthermore, all traditional vegetation indices have a vital uncertainty, that allow for the determination of the threshold of VIs. Fortunately, based on the theory of being above the baseline, one advantage of using the MFI in detecting mangrove forests is that the threshold is at the fixed value of zero.

This study provides an index built by Sentinel-2 MSI bands for discriminating submerged mangrove forests from water background. It supports the findings of previous studies that the NIR and red-edge provide great opportunities in discriminating between vegetation and water. Sentinel-2 MSI image contains five bands in NIR region, four of which were used to build the MFI. The FAI was also established based on baseline theory, but with one NIR band. However, according to Figure 7 and our statistics, unlike MFI (completely separated submerged mangrove forest and water), in the FAI image, 5% of submerged mangrove forests pixels were mixed with water pixels. This is primarily because unexpected fluctuation in one NIR band could greatly affect the value of the FAI. The four MSI red-edge bands demonstrated relatively stable discrimination between submerged mangrove forest and water.

Theoretically, the MFI concept can be applied to other sensors that contain spectral channels of red, NIR, and SWIR, for example, the Landsat OLI sensor which has a red band ranging from 630 to 690 nm, an NIR band ranging from 840 to 890 nm, and a SWIR band ranging from 2100 to 2300 nm. However, different sensors may acquire different MFI values due to the different ranges of red, NIR, and SWIR bands. Furthermore, the performance of the Sentinel-2 MSI red-edge bands will also be present on the Sentinel-3 Ocean and Land Color Instrument (OLCI) sensor [51]. Therefore, the adaptability of the MFI to other remote sensing sensors still requires further examination. Moreover, the MFI was designed based on the reflectance peak in the NIR spectral regions of green vegetation. Therefore,

the MFI has great potential in detecting any submerged or emerged vegetation in aquatic environments, such as floating algae and aquatic macrophytes. However, considering the various environments where aquatic vegetation grows, the applicability of the MFI in detecting other aquatic vegetation still warrants further exploration.

### 4.2. Uncertainties Leading to Overestimation of Mangrove Forests Using the MFI

This study demonstrates that there are abundant differences in the spectral reflectance between submerged mangrove forests and water bodies. In addition, the spectral curves of submerged and emerged mangrove forests showed similar concave–convex characteristics (Figure 5). Therefore, the MFI function can efficiently identify and detect mangrove forests from water background. However, the MFI was designed based on the reflectance peak between red and SWIR; any other vegetation that contains absorption signatures of chlorophyll in aquatic environment can also be detected [17,36,52]. Hence, pixels containing floating vegetation (for example, algae) and other aquatic macrophytes (for example, *Spartina alterniflora*) may be classified as mangrove forest. Additionally, due to limits in image resolution, a small area of the water could still be classified as mangrove forests, due to having similar spectral characteristics as nearby mangrove forests. These uncertainties can lead to overestimation of mangrove forests by the MFI. In our application in Dalhousie Island, Sundarbans, India, the bias of area of mangrove forests obtained by MFI is 2% larger.

### 4.3. Limitations Leading to Underestimation of Mangrove Forests Using the MFI

In this study, MFI was created based on the typical reflectance curves of submerged mangrove forests. However, the spectral curve of submerged mangrove forest can be affected by several factors, including water transparency (turbidity), distance that mangrove canopy under the water surface, and the coverage of mangrove forest [15]. Liew and Chang proved that the spectral curves of submerged vegetation changes when water turbidity and depth change [53]. They demonstrated that with high turbidity (50 nephelometric turbidity units), green vegetation could not be distinguished at a water depth of 0.5 m. In addition, with low turbidity (0.5 nephelometric turbidity units), typical vegetation reflectance was undetectable at a water depth of 1 m. Chen et al. discovered that when submerged vegetation coverage was less than 40%, it is difficult to detect vegetation based on the NIR peak in the spectral reflectance curve [15]. Unfortunately, water flow in mangrove swamps always has high turbidity, and newly grown trees at the edge of mangrove forests always have low coverage. According to our field measurement, in Zhenzhu Harbor, submerged mangrove forests with a depth of 60 cm under the water surface would not be detected by MFI. Moreover, due to limits of image resolution, small parts of the mangrove forests may be classified as water due to low tree coverage. As shown in Figure 12, in Zhenzhu Harbor, low mangrove forests along tidal creeks were not identified by MFI. These limitations lead to underestimation of the areal extent of mangrove forests by the MFI.

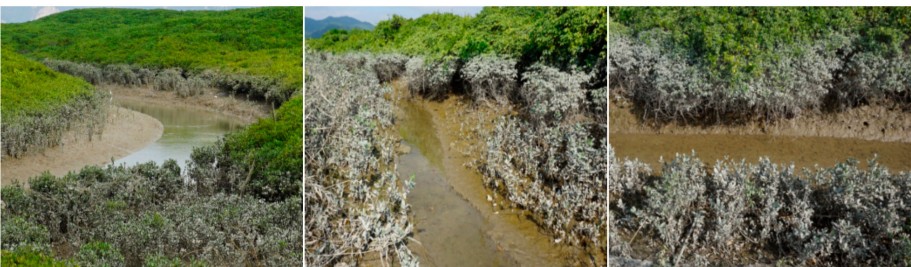

**Figure 12.** Intermittently flooded mangrove forests in local low tide period.

## 5. Conclusions

Based on the spectral response curves of submerged mangrove forests, a new vegetation index (MFI) was developed to distinguish mangrove forests from the water background. To take full

advantage of the differences in reflectance between submerged mangrove forests and the water background, Sentinel-2 MSI bands, red and SWIR2 were selected to build a linear baseline, and the average reflectance value of four red-edge bands above the baseline was defined as mangrove forest index (MFI). This new vegetation index is more advantageous in detecting submerged mangrove forests than the traditional NDVI, LSWI, MNDWI, and FAI indices. According to the results of Jensen–Shannon divergence, MFI significantly widens the distance of submerged mangrove forest and water pixels compared to other VIs (the Jensen-Shannon divergence values of MFI, FAI, NDVI, LSWI, and MNDWI are 0.209, 0.077, 0.012, 0.003, and 0.121, respectively). Theoretically, 100% of submerged mangrove forests could be extracted from MFI images. Practically, application of the MFI in three global mangrove sites showed 99% to 102% of submerged mangrove forests were successfully extracted from the MFI image. The overall accuracy of classification results obtained from the MFI image ranged from 91.7% to 97.6%. According to our field measurements in Zhenzhu Harbor, MFI is insensitive to mangrove forests with canopies under 60 cm of the water surface. There are some uncertainties and limitations, but the MFI was proven to be effective in detecting the extent and condition of mangrove forests from high-tide Sentinel MSI images. Although the repeatability and portability of the MFI is still a work in progress, this index brings great benefits to remote sensing communities of coastal and aquatic vegetation studies.

**Author Contributions:** M.J. and Z.W. designed the research, process the data, and wrote the manuscript draft. Y.Z. helped with designed research and reviewed the manuscript. C.W. helped with image analysis, fieldwork, and reviewed the manuscript. D.M. helped with image analysis and reviewed the manuscript.

**Funding:** The work is supported by Science and Technology Basic Resources Investigation Program of China (No. 2017FY100706), the National Natural Science Foundation of China (No. 41601470, No. 41601406), the Strategic Planning Project of the Institute of Northeast Geography and Agroecology (IGA), Chinese Academy of Sciences (No. Y6H2091000), and the Youth Innovation Promotion Association of Chinese Academy of Sciences (2017277, 2012178). This work is supported by Open Fund of State Laboratory of Information Engineering in Surveying, Mapping and Remote Sensing, Wuhan University (Grant No. 19I02).

**Acknowledgments:** The authors are grateful to the colleagues who participated in the field surveys and data collection.

**Conflicts of Interest:** The authors declare no conflict of interest.

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
