# Peer review of "A New Vegetation Index to Detect Periodically Submerged Mangrove Forest Using Single-Tide Sentinel-2 Imagery"

_remotesensing, doi:10.3390/rs11172043_

Round 1

Reviewer 1 Report

p.p1 {margin: 0.0px 0.0px 0.0px 0.0px; font: 12.0px Helvetica}

The paper proposes a new vegetation index to map the distribution of underwater mangrove using Sentinal-2 image. As water level along the coast and river channels tend to affect by the tide, mapping submerged mangrove is a sound idea in the area of mangrove research and useful to locate the pioneer species. The idea of using the bands to formulate the MFI is rational and sound though the rationale is quite similar to the floating vegetation index (FVI) in another article. As Sentinal-2 data is free, the merit of the proposed index is widely applicable. More comments are shown in the attachment.

Reviewer 2 Report

In this manuscript, the authors proposed a new vegetation index (Mangrove Forest Index; MFI) for the detection of mangrove forests using Sentinel-2 satellite imagery. My comments are listed as below.

First, this new vegetation index, MFI, seems interesting with respect to the detection of mangrove forests particularly for those that are periodically submerged due to tidal effect. However, I didn’t see a strong comparison between this index with other commonly used indices for the need of using this new index for the detection of mangrove forest. Based on the results of vegetation index reported in this study (section 3.2), it seems other vegetation indices are also applicable for the detection of mangrove forest. Further, section 3.1 (definition of the new index) needs to be in Section 2.

Second, the use of accuracy metric to evaluate the classification performance is only based on quantity (area) instead of spatial configuration (e.g., based on confusion matrix). It is hard to tell the utility of this new vegetation index from the results (in Section 3.3) reported in this study based on the accuracy metric used.

Third, Discussion and Conclusion sections are weak and need more work.

Fourth, the English language of this manuscript need to be improved. For example, many long sentences were used.

Minor comments:

1)     Line 33: “… accurately mapping of mangrove forests…”: drop “of”

2)     Line 136-138, “In this study…, workflow is shown as Figure 2A” this sentence needs to be rephrased.

3)     Line 191, “Existed”-> Existing

4)     Line 285, Indian->India

Reviewer 3 Report

Comments to the authors

It is a novel method to derive submerged vegetation, which would contribute to high resolution monitoring of coastal vegetation worldwide. Especially, Subsection 4.3 highlights the advantage of the study. However, the evaluation of the new index is arbitrary and insufficient. With an appropriate statistical measure, the study can be more significant and reliable.

C merely measures difference of the locations (i.e. means) of two distributions but does not take the scale (i.e. variation) of each distribution into account. Arbitrary scaling of the distributions (L237-239) results in an arbitrary C value, which is not comparable with the other Cs. Moreover, the lack of the distribution scale in C might be a reason why FAI had higher C in spite of the more confusion (L243-244). Use a statistical distance indicator that consider both the location and the scale, such as the Jensen-Shannon divergence, instead of C, and you can compare the performance of the indices directly and objectively without any arbitrary and subjective processing like L237-239.

L80-88       Here you mention only the normalized difference indices and their limitations. It is suggested, however, to also mention the baseline-based indices and their limitations from which you have got the idea of your study.

L85-88       It is suggested to stress another limitation of the normalized difference indices that the threshold is not certain as you point out at L343-344. By presenting the limitations of both the normalized difference and the baseline-based indices, you will be able to state the objective of your study more clearly (L89-91).

L91            “imagery” is a mass noun.

L110-111   How did you resample the pixels? Averaging? Sampling?

L119-120   Full genus names are required at the first citation.

L129 (Figure 1) Captions/explanations for the maps are also required.

L154 (Figure 2) These two figures can be separated with more detailed captions.

L161           The full name for the abbreviation “NIR” is required at the first use. So is for “SWIR”, too.

L173-174   “… and submerged mangrove forest.” -> “… and submerged mangrove forest that are observed on the MSI image”

L174-180 & Figure 4 Give symbol letters to each of the landcover types (not only the submerged (a) & (b)) so that they can be more easily identified both in the text and on Figure 4.

L188 (Figure 4) Indicate the location (wavelength) of the MSI bands on the figure. Remove the Baseline from the figure, then create another figure with the Baseline when explaining the idea of MFI, because you have not yet described the idea of the baseline at 2.4.

L191-212   Since it is under the Material and Methods section, explain only the indices that you used in the analysis. Which bands of MSI are used to derive each index? See comments for L80-88 and L85-88 for the general examination of the existing indices.

L191           “index” -> “indices”

L192           “Formula” -> “Equation”

L213-218   As pointed out above, C is not an appropriate measure for the comparison. Consider a statistical distance indicator instead.

L220-233   Move this part to somewhere between 2.4 & 2.5 or 2.5 & 2.6, because it is not a result but the core of the method of this study. Create a new figure that explain the idea of MFI graphically. See comment for Figure 4.

L234-247   As pointed out above, C did not work as an objective measure and you had to rely on subjective interpretations here. Consider a statistical distance indicator instead.

L249 (Figure 5) Explain the elements of the boxplot.

L249 (Figure 5) MNDWI seems to have a comparable performance with MFI. It is suggested to mention this in the discussion and stress the advantage of MFI that its threshold has been fixed at zero by definition while MNDWI’s has not.

L255 (Table 3) Check values such as Max_Submerged of MFI, Min_Submerged of NDVI and Mean_Water of NDVI. What do capitalized values indicate?

L257           “Global applications of MFI” Instead of this title, how about “Visual evaluation of MFI at different mangrove forests around the world”?

L258           “randomly selected” not randomly at all!

L261           “Indian” -> “India”

L266-278   How did you delineate the mangrove forests, both submerged and emerged, from other terrestrial landcovers?

L343-344   Show the evidence by referring Figure 5 and the related text. See comment for Figure 5.

L348-350   Show the evidence derived from the study (by statistical distance indicator).

L349           “Sentinel-2A” -> “Sentinel-2” ???

L363           “MFI” the full name is necessary here.

L368           “NDVI, LSWI, and MNDWI” Why is FAI excluded?

Abstract can be revised in accordance with the text revision.